# Genome-wide association study identifies genetic loci for self-reported habitual sleep duration supported by accelerometer-derived estimates

Hassan S. Dashti [1,2], Samuel E. Jones [3], Andrew R. Wood et al.[#]

Sleep is an essential state of decreased activity and alertness but molecular factors regulating sleep duration remain unknown. Through genome-wide association analysis in 446,118 adults of European ancestry from the UK Biobank, we identify 78 loci for self-reported habitual sleep duration ($p < 5 \times 10^{-8}$; 43 loci at $p < 6 \times 10^{-9}$). Replication is observed for *PAX8*, *VRK2*, and *FBXL12/UBL5/PIN1* loci in the CHARGE study ($n = 47,180$; $p < 6.3 \times 10^{-4}$), and 55 signals show sign-concordant effects. The 78 loci further associate with accelerometer-derived sleep duration, daytime inactivity, sleep efficiency and number of sleep bouts in secondary analysis ($n = 85,499$). Loci are enriched for pathways including striatum and subpallium development, mechanosensory response, dopamine binding, synaptic neurotransmission and plasticity, among others. Genetic correlation indicates shared links with anthropometric, cognitive, metabolic, and psychiatric traits and two-sample Mendelian randomization highlights a bidirectional causal link with schizophrenia. This work provides insights into the genetic basis for inter-individual variation in sleep duration implicating multiple biological pathways.

Sleep is an essential homeostatically regulated state of decreased activity and alertness conserved across animal species, and both short and long sleep duration associate with chronic disease and all-cause mortality[1,2]. Research in model organisms (reviewed in refs. [3,4]) has delineated aspects of the neural circuitry of sleep–wake regulation[5] and molecular components including specific neurotransmitter and neuropeptide systems, intracellular signaling molecules, ion channels, circadian clock genes and metabolic and immune factors[4], and more recently phosphorylation of synaptic proteins[6], but their specific roles and relevance to human sleep regulation are largely unknown. Prospective epidemiologic studies suggest that both short (<6 h per night) and long (>9 h per night) habitual self-reported sleep duration associate with cognitive and psychiatric, metabolic, cardiovascular, and immunological dysfunction as well as all-cause mortality compared to sleeping 7–8 h per night[7–9]. Furthermore, chronic sleep deprivation in modern society may lead to increased errors and accidents[10]. Yet, whether short or long habitual sleep duration causally contributes to disease initiation or progression remains to be established.

Habitual self-reported sleep duration is a complex trait with an established genetic component (twin- and family-based heritability ($h^2$) estimates = 9–45%[11–14]). Candidate gene sequencing in pedigrees and functional validation of rare, missense variants established *BHLHE41* (previously *DEC2*), a repressor of *CLOCK/ARNTL* activity, as a causal gene[15,16], supporting the role of the circadian clock in sleep regulation. Previous genome-wide association studies (GWASs), including a recent GWAS in up to 128,286 individuals, identified association of common variants at or near the *PAX8* and *VRK2* genes, among other signals that have not yet been replicated[13,14,17–19].

Here, we extend GWAS of self-reported sleep duration in UK Biobank, test for consistency of effects in independent studies of adults and children/adolescents, determine their impact on accelerometer-derived estimates, perform pathway and tissue enrichment to highlight relevant biological processes, and explore causal relationships with disease traits.

## Results

**GWAS for self-reported habitual sleep duration.** Among UK Biobank participants of European ancestry ($n = 446,118$), mean self-reported habitual sleep duration was 7.2 h (1.1 standard deviation) per day (Supplementary Table 1). GWAS using 14,661,600 imputed genetic variants identified 78 loci ($P < 5 \times 10^{-8}$; Fig. 1a, Supplementary Data 1,2, Supplementary Figure 1a). Individual signals exert an average effect of 1.04 min (0.34 standard deviation) per allele, with the largest effect at the *PAX8* locus, with an estimate of 2.44 min (0.16 standard error) per allele. The 5% of participants carrying the most sleep duration-increasing alleles self-reported 22.2 min longer sleep duration compared to the 5% carrying the fewest. The 78 loci explained 0.69% of the variance in sleep duration, and genome-wide single-nucleotide polymorphism (SNP)-based heritability was estimated at 9.8 (0.1)%. Of the 78 variants, 43 variants passed a more stringent multiple correction threshold of $P < 6 \times 10^{-9}$ established by permutation testing for a related sleep trait[20].

Sensitivity analyses indicated that the 78 genetic associations were largely independent of known risk factors (Supplementary Data 3). Effect estimates at 15/78 loci were attenuated by 15–25% upon adjustment for frequent insomnia symptoms, perhaps reflecting contribution to an insomnia sub-phenotype with physiological hyperarousal and objective short sleep duration[21] (Supplementary Data 3). Effect estimates at 19/78 were also slightly attenuated after adjustment of lifestyle factors. No signal attenuation was observed when accounting for body mass index

(BMI) at rs9940646 at *FTO*, the established BMI-associated signal ($r^2 = 0.81$ with rs9939609[22] and where the higher BMI allele associated with shorter sleep duration). Analysis conditioned on the lead SNPs in each genomic region identified 4 secondary association signals at the *VRK2*, *DAT1 (SLC6A3)*, *DRD2*, and *MAPT* loci (Supplementary Table 2). Effect estimates were largely consistent in GWAS excluding shift workers and those with prevalent chronic and psychiatric disorders (excluding $n = 119,894$ participants) (Supplementary Data 1, 2, Supplementary Table 3, Supplementary Figure 1b, 2). GWAS results were similar for men and women ($r_g$ (SE) = 0.989 (0.042); $P < 0.001$) (Supplementary Table 4, Supplementary Figure 1c, 1d, 3).

**GWAS for self-reported short and long sleep.** Separate GWAS for short (<7 h; $n = 106,192$ cases) and long (≥9 h; $n = 34,184$ cases) sleep relative to 7–8 h sleep duration ($n = 305,742$ controls) highlighted 27 and 8 loci, respectively, of which 13 were independent from the 78 sleep duration loci (Fig. 1b, Supplementary Data 2,4, Supplementary Table 5, Supplementary Figures 1e, 1f). Only the *PAX8* signal was shared across all three traits, consistently indicating associations between the minor allele and longer sleep duration. For most long sleep loci, we could exclude equivalent effects on short sleep based on 95% confidence intervals (CIs) of effect estimates (Supplementary Figure 4, Supplementary Table 5). Sensitivity analyses accounting for factors potentially influencing sleep did not alter the results (Supplementary Data 5, Supplementary Table 6).

**Replication of sleep duration loci in independent studies.** We tested for independent replication of lead loci in the CHARGE (Cohorts for Heart and Aging Research in Genomic Epidemiology) consortium GWAS of adult sleep duration ($n = 47,180$ from 18 studies[14]) and observed replication evidence for individual association signals at the *PAX8*, *VRK2*, and *FBXL12/UBL5/PIN1* loci ($P < 6.4 \times 10^{-4}$; Supplementary Data 2,6,7, Supplementary Figure 5a), and nominal replication ($P < 0.05$) for 14 additional loci. Of the 70 loci covered in the CHARGE consortium, 55 signals showed a consistent direction of effect (binomial $P = 6.1 \times 10^{-7}$), and a combined weighted genetic risk score (GRS) of the 70 signals was associated with a 0.66 min (95% CI: 0.54–0.78) longer sleep per allele ($P = 1.23 \times 10^{-25}$) in the CHARGE consortium (Table 1). Consistently strong genetic correlation was observed between the CHARGE consortium and UK Biobank studies ($r_g$ (SE) = 1.00 (0.123); $P < 0.001$; Supplementary Table 7). In meta-analysis of CHARGE consortium and UK Biobank studies, 52/70 signals retained GWAS significance, and 38/70 signals passed the more stringent multiple correction threshold of $P < 6 \times 10^{-9}$ (Supplementary Data 6).

In the childhood/adolescent GWAS for sleep duration from the EAGLE (EArly Genetics and Lifecourse Epidemiology) consortium[19] ($n = 10,554$), none of the 78 GWAS signals showed independent replication (all $P > 0.05$; Supplementary Data 6, 7, Supplementary Figure 5b). Of the 77 loci covered in the EAGLE consortium, marginal evidence of association was observed for the adult sleep duration loci, with 45/77 signals demonstrating consistent directionality (binomial $P = 0.031$). For a combined 77 SNP GRS, we observed an effect of 0.16 min (95% CI: 0.02–0.30) longer sleep per allele ($P = 0.03$; Table 1). No significant overall genetic correlation was observed with GWAS of adult sleep duration ($r_g$ (SE) = 0.098 (0.076), $P = 0.20$ with UK Biobank; Supplementary Table 7). In meta-analysis of all three sleep duration GWASs, 56/78 signals retained GWAS significance, and 40/78 passed the more stringent multiple correction threshold of $P < 6 \times 10^{-9}$ (Supplementary Data 2, 6, 7, Supplementary Figure 5c).

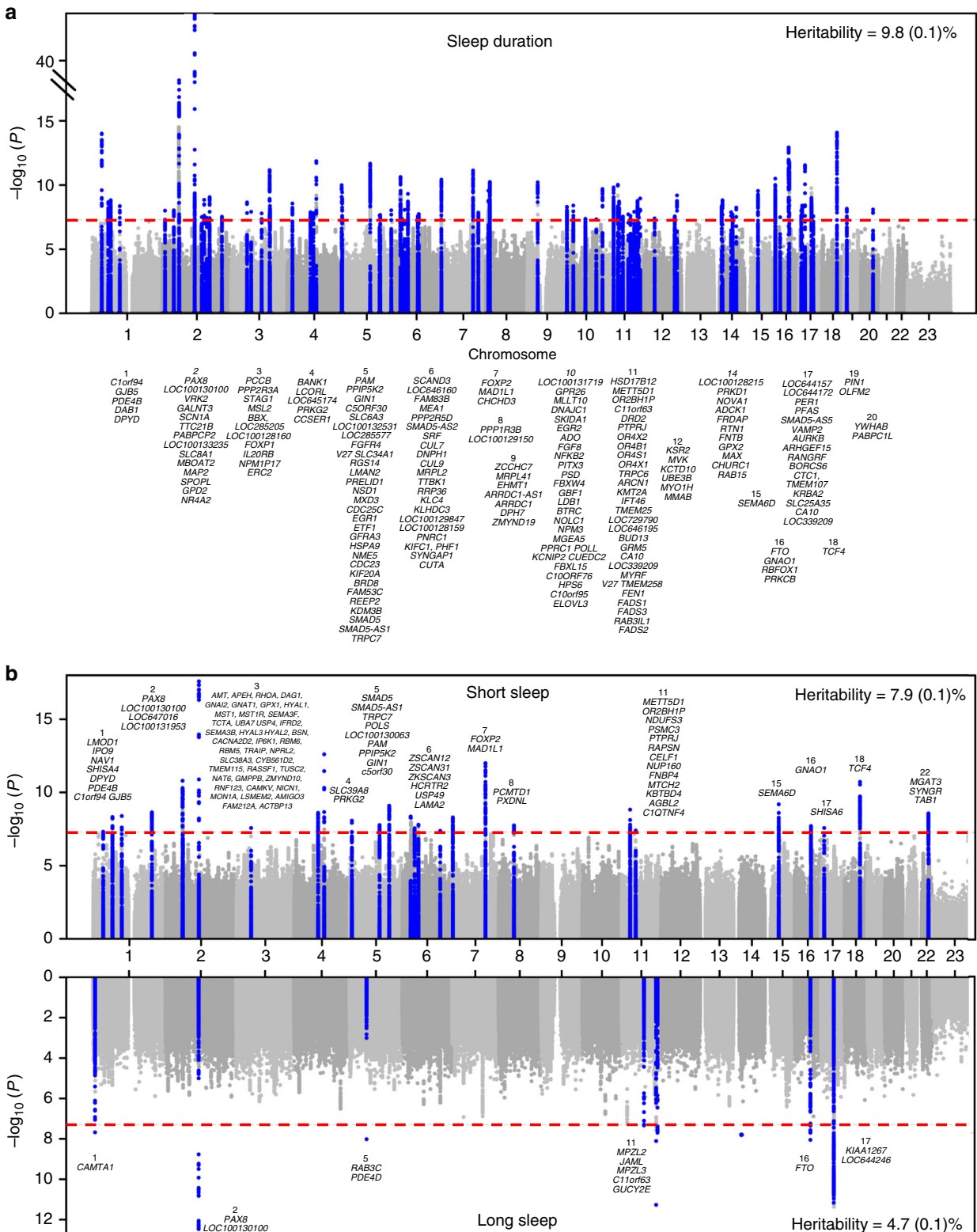

**Fig. 1** Plots for genome-wide association analysis results for sleep duration and short/long sleep. **a** Manhattan plot of sleep duration ($n = 446{,}118$) and **b** Miami plot of short (cases $n = 106{,}192/305{,}742$) and long (cases $n = 34{,}184/305{,}742$) sleep. Plots show the $-\log_{10}P$ values ($y$-axis) for all genotyped and imputed single-nucleotide polymorphisms (SNPs) passing quality control in each genome-wide association study (GWAS), plotted by chromosome ($x$-axis). Blue peaks represent genome-wide significant loci. Horizontal red line denotes genome-wide significance ($P = 5 \times 10^{-8}$)

**Table 1 A risk score of genetic variants for self-reported sleep duration (78 SNPs), self-reported short (27 SNPs) or long (8 SNPs) sleep duration associates with self-reported sleep duration in the CHARGE (adult) consortium (n = 47,180), self-reported sleep duration in the EAGLE (childhood/adolescent) consortium (n = 10,554), and activity-monitor-based measures of sleep fragmentation, timing, and duration from 7-day accelerometry in the UK Biobank (n = 85,499)**

| Trait | Sleep duration GRS | | Short sleep GRS | | Long sleep GRS | |
|---|---|---|---|---|---|---|
| | Beta or OR* (95% CI) per effect allele | P value | Beta or OR* (95% CI) per effect allele | P value | Beta or OR* (95% CI) per effect allele | P value |
| CHARGE Study (n = 47,180); self-reported sleep duration (min)[a] | **0.66 (0.54 to 0.78)** | **$1.23 \times 10^{-25}$** | | | | |
| EAGLE Study (n = 10,554); self-reported sleep duration (min)[b] | **0.16 (0.02 to 0.30)** | **$2.80 \times 10^{-2}$** | | | | |
| UK Biobank 7-day accelerometry (n = 85,499) sleep duration estimates | | | | | | |
| Sleep duration (min) | **0.47 (0.40 to 0.53)** | **$1.93 \times 10^{-44}$** | **−0.43 (−0.56 to −0.31)** | **$1.21 \times 10^{-11}$** | **2.12 (1.65 to 2.59)** | **$1.08 \times 10^{-18}$** |
| Short sleep duration (n = 13,760 cases, 66,110 controls) | **0.98 (0.98 to 0.99)*** | **$4.00 \times 10^{-19}$** | **1.02 (1.01 to 1.02)*** | **$4.91 \times 10^{-6}$** | **0.94 (0.92 to 0.97)*** | **$1.10 \times 10^{-5}$** |
| Long sleep duration (n = 5629 cases, 66,110 controls) | **1.01 (1.01 to 1.02)*** | **$3.78 \times 10^{-9}$** | 0.99 (0.98 to 1.00)* | 0.11 | **1.10 (1.07 to 1.14)*** | **$1.29 \times 10^{-8}$** |
| Daytime inactivity duration (min) | **0.08 (0.03 to 0.13)** | **$2.74 \times 10^{-3}$** | 0.01 (−0.09 to 0.11) | 0.89 | **0.65 (0.28 to 1.02)** | **$6.49 \times 10^{-4}$** |
| Sleep duration standard deviation (min) | −0.02 (−0.07 to 0.02) | 0.34 | 0.05 (−0.04 to 0.14) | 0.26 | −0.07 (−0.40 to 0.27) | 0.69 |
| Sleep fragmentation estimates | | | | | | |
| Sleep efficiency % | **0.05 (0.04 to 0.06)** | **$8.38 \times 10^{-23}$** | **−0.05 (−0.07 to −0.04)** | **$4.79 \times 10^{-9}$** | **0.15 (0.08 to 0.22)** | **$1.56 \times 10^{-5}$** |
| Number of sleep bouts (n) | **0.02 (0.01 to 0.02)** | **$1.59 \times 10^{-10}$** | **−0.01 (−0.02 to 0.00)** | **$2.42 \times 10^{-3}$** | 0.02 (−0.01 to 0.05) | 0.24 |
| Sleep timing estimates | | | | | | |
| Midpoint of 5 h daily period of minimum activity (L5 timing) (minutes) | −0.05 (−0.13 to 0.03) | 0.23 | 0.07 (−0.09 to 0.22) | 0.41 | 0.39 (−0.20 to 0.97) | 0.20 |
| Midpoint of 10 h daily period of maximum activity (M10 timing) (minutes) | 0.03 (−0.06 to 0.12) | 0.51 | −0.05 (−0.23 to 0.12) | 0.55 | 0.65 (−0.02 to 1.32) | $6.00 \times 10^{-2}$ |
| Sleep midpoint (min) | −0.03 (−0.07 to 0.01) | 0.20 | 0.01 (−0.07 to 0.08) | 0.88 | 0.05 (−0.24 to 0.34) | 0.74 |

Genetic risk scores for sleep duration, short sleep and long sleep were tested using the weighted genetic risk score calculated by summing the products of the sleep trait risk allele count for all 78, 27, or 8 genome-wide significant SNPs multiplied by the scaled effect from the primary genome-wide association study (GWAS) using the GTX package in R. Effect estimates (beta or OR) are reported per additional effect allele for sleep duration, short sleep, or long sleep. Significant GRS associations (P < 0.05) are shown in bold.
SNP single-nucleotide polymorphism, CI confidence interval, GRS genetic risk score, OR odds ratio, CHARGE Cohorts for Heart and Aging Research in Genomic Epidemiology, EAGLE EArly Genetics and Lifecourse Epidemiology
[a]Self-reported and varied by cohorts, typically: "How many hours of sleep do you usually get at night (or your main sleep period)?"
[b]In all cohorts, except in GLAKU, child sleep duration was assessed by a single, parent-rated, open question, "How many hours does your child sleep per day including naps?" In GLAKU, parents were asked about the usual bed and rise times during school days, from which the total sleep duration could be estimated
*indicates OR (95% CI)

**Association of sleep duration loci with objective sleep.** Given the limitations of self-reported sleep duration[23,24], and in order to explore underlying physiologic mechanisms, in secondary analyses, we tested the 78 lead variants for association with 8 accelerometer-derived sleep estimates in a subgroup who had completed up to 7 days of wrist-worn accelerometry (n = 85,499; Supplementary Table 8)[25]. The lead PAX8 genetic variant was associated with 2.68 min (0.29) longer sleep duration (compared to 2.44 min (0.16) by self-report), 0.21% (0.04%) greater sleep efficiency, and 0.94 min (0.23) greater daytime inactivity duration per minor A allele (P < 0.00064; Supplementary Data 8). The 5% of participants carrying the most sleep duration-increasing alleles were estimated to have 9.7 min (95% CI: 7.5–11.8) accelerometer measured longer sleep duration compared to the 5% carrying the fewest. The 78 SNP GRS associated with longer accelerometer-derived sleep duration, longer duration of daytime inactivity, greater sleep efficiency, and larger number of sleep bouts, but not with day-to-day variability in sleep duration or estimates of sleep timing (Table 1). A GRS of 27 short sleep variants was associated with shorter accelerometer-derived sleep duration, lower sleep efficiency, and fewer sleep bouts, whereas a GRS of 8 long sleep variants associated with longer accelerometer-derived sleep duration, higher sleep efficiency, and longer daytime inactivity (Table 1, Supplementary Data 9).

**Functional annotation for identified loci.** The sleep duration association signals encompass >200 candidate causal genes determined by SNPsea[26] through assessing linkage disequilibrium (LD) intervals of each identified loci, defined by the furthest SNPs in a 1 Mb window with $r^2 > 0.05$, and a summary of reported gene–phenotype annotations is shown in Supplementary Data 10. Compelling candidates include genes in the dopaminergic (DRD2, SLC6A3), MAPK/ERK (mitogen-activated protein kinase/ extracellular signal-regulated kinase) signaling (ERBB4, VRK2, KSR2), orexin receptor (HCRTR2), and GABA (GABRR1) signaling systems[4,27]. Further, studies of sleep regulation in animal models prioritize several candidates (GABRR1, GNAO1, HCRTR2, NOVA1, PITX3, SLC6A3, DRD2, and VAMP2 for sleep duration; PDE4B and SEMA3F for short sleep; PDE4D for long sleep). Circadian genes within associated loci include PER3, BTRC, and the previously implicated PER1[28], which may act through glucocorticoid stress-related pathways to influence sleep duration. Association signals at 4 loci directly overlapped with other GWAS signals ($r^2 > 0.8$ in 1KG CEU; from the National Human Genome Research Institute (NHGRI)), with the shorter sleep allele associated with higher BMI (FTO), increased risk of Crohn's disease (NFKB1, SLC39A8, BANK1 region), febrile seizures and generalized epilepsy (SCN1A), and cardiometabolic risk (FADS1/2 gene cluster), and decreased risk of interstitial lung disease (MAPT/KANSL). Fine-mapping using credible set analysis in PICS[29] highlighted 52 variants (Supplementary Data 11, 12). Partitioning of heritability by functional annotations identified excess heritability across genomic regions conserved in mammals, consistent with earlier findings[18], and additionally in regions with active promoters and enhancer chromatin marks (Supplementary Data 13).

**Gene- and pathway-based analysis.** Gene-based tests identified 235, 54, and 20 genes associated with sleep duration, short sleep, and long sleep, respectively ($P \leq 2.29 \times 10^{-6}$; Supplementary

Data 14, 15). Pathway analyses of these genes using MAGMA[30] and Pascal[31] indicated enrichment of pathways including striatum and subpallium development, mechanosensory response, dopamine binding, catecholamine production, and long-term depression (Fig. 2a, b, Supplementary Table 9, 10). In agreement with the *FADS1/2* signal, we also observe enrichment in genes related to unsaturated fatty acid metabolism. A custom pathway analysis in Pascal indicated enrichment of association in a gene-set of synaptic sleep-need-index phosphoproteins (SNIPPs), which have recently been demonstrated to be differentially phosphorylated based on sleep need in mouse models[6] (Supplementary Table 11; $P_{emp} = 1.44 \times 10^{-4}$). Of these, associations at *SCN1A* and *PDE4B* are genome-wide significant.

Tissue enrichment analyses of gene expression from Genotype-Tissue Expression (GTEx) tissues identified enrichment of associated genes in several brain regions including the cerebellum, a region of emerging importance in sleep/wake regulation[32], frontal cortex, anterior cingulated cortex, nucleus accumbens, caudate nucleus, hippocampus, hypothalamus, putamen, and amygdala (Fig. 2c, Supplementary Table 12). Enrichment was also

observed in the pituitary gland. Single cell enrichment analyses in FUMA using a recently described Tabula Muris[33] dataset showed enrichment in brain neurons and pancreatic alpha cells (Supplementary Data 16). Integration of gene expression data with GWAS using transcriptome-wide association analyses in 11 tissues[34] identified 38 genes for which sleep duration SNPs influence gene expression in the tissues of interest (Supplementary Data 17).

Several lead SNPs were associated with one or more of 3144 human brain structure and function traits assessed in the UK Biobank ($P < 2.8 \times 10^{-7}$, $n = 9,707$; Oxford Brain Imaging Genetics Server;[35] Supplementary Figure 6). These include associations between the *PAX8* locus with resting-state functional magnetic resonance imaging networks (Supplementary Figure 6a, 6h, 6m), rs13109404 (*BANK1*; Supplementary Figure 6b) and bilateral putamen and striatum volume, possibly relating to functional findings on reward processing after experimental sleep deprivation[36], and rs330088 (*PPP1R3B* region; Supplementary Figure 6c) and temporal cortex morphometry, which may relate to recent findings showing extreme sleep durations predict subsequent frontotemporal gray matter atrophy[37].

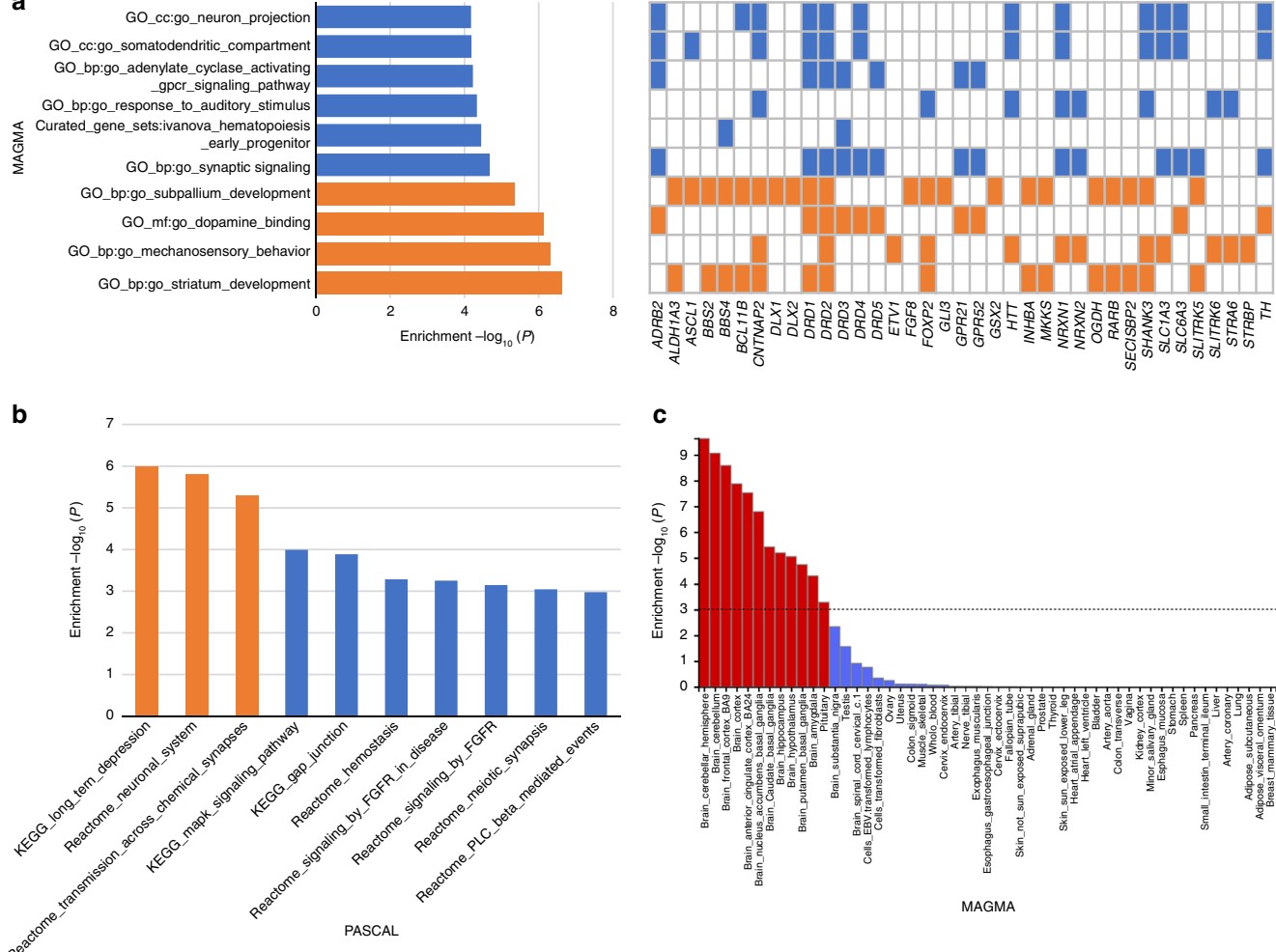

**Fig. 2** Pathway-based and tissue expression enrichment analyses for sleep duration. **a** Pathway analysis is based on MAGMA gene sets. Top 10 of 10,891 pathways are shown, and significant pathways are indicated in orange ($P < 4.59 \times 10^{-6}$). For each significant pathway, respective sleep genes are indicated with a colored orange box. Sleep genes from significant pathways that overlap with remaining pathways are indicated in blue. **b** Pathway analysis is based on Pascal (gene-set enrichment analysis using 1077 pathways from KEGG, REACTOME, BIOCARTA databases). Top 10 pathways are shown, and significant pathways are indicated in orange ($P < 4.64 \times 10^{-5}$). **c** MAGMA tissue expression analysis using gene expression per tissue based on GTEx RNA-seq data for 53 specific tissue types. Significant tissues are shown in red ($P < 9.43 \times 10^{-4}$). All pathway and tissue expression analyses in this figure can be found in tabular form in Supplementary Tables 9, 10, 12

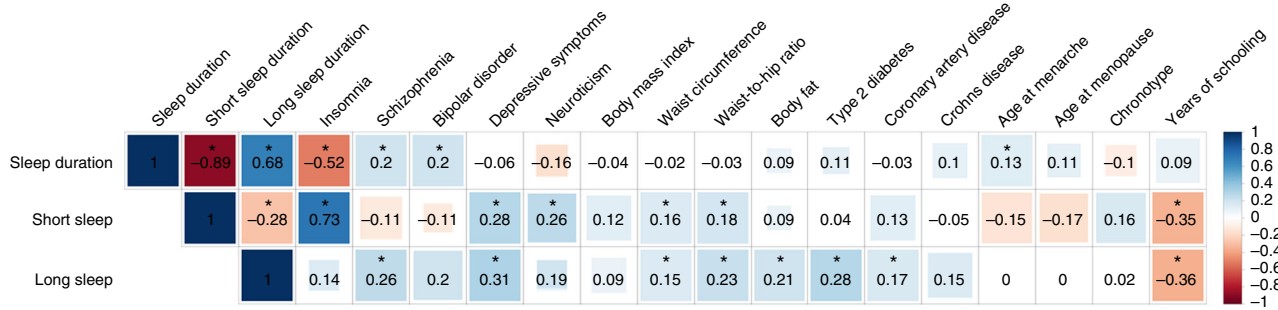

**Fig. 3** Shared genetic architecture between sleep duration and behavioral and disease traits. Linkage disequilibrium (LD) score regression estimates of genetic correlation ($r_g$) were obtained by comparing genome-wide association estimates for sleep duration with summary statistics estimates from 224 publically available genome-wide association studies (GWASs). Blue indicates positive genetic correlation and red indicates negative genetic correlation; $r_g$ values are displayed for significant correlations. Larger colored squares correspond to more significant $P$ values, and asterisks indicate significant ($P < 2.2 \times 10^{-4}$) genetic correlations. All genetic correlations in this report can be found in tabular form in Supplementary Data 18

**Genetic correlation and Mendelian randomization**. Genome-wide genetic correlations using LD score regression analyses[38] indicated shared links between sleep duration and eight cognitive, psychiatric, and physical disease traits (Fig. 3, Supplementary Data 18). We observed modest positive genetic correlations between sleep duration and schizophrenia, bipolar disorder, and age at menarche, and a negative correlation with insomnia that persisted even upon excluding participants with psychiatric disorders, indicating that genetic relationships are not driven by the presence of co-morbid conditions. In addition, both short and long sleep showed positive genetic correlations with depressive symptoms, waist circumference, and waist-to-hip ratio, and negative correlations with years of schooling. For short sleep, genetic correlations were also observed with insomnia, neuroticism, and smoking, and for long sleep, positive correlations were evident with schizophrenia, body fat, type 2 diabetes, and coronary artery disease.

Mendelian randomization (MR) analyses to test for causal links between sleep duration and genetically correlated traits suggested longer sleep duration is causal for increased risk of schizophrenia (two-sample MR: inverse variance weighted: 0.0088 (0.003) log odds ratio per min, $P = 3.70 \times 10^{-3}$; weighted median: 0.008 (0.003) log odds ratio per min, $P = 3.95 \times 10^{-3}$) (Fig. 4, Supplementary Table 13). These data suggest that a 1 h longer sleep duration leads to a 69.6% increase in the risk for schizophrenia. In leave-one-out sensitivity analyses, MR results remained robust and consistent (all $P < 6.86 \times 10^{-3}$; Supplementary Data 19). Sensitivity MR analyses limited to signals from GWAS adjusting for confounders (BMI, insomnia, or other lifestyle traits) and using corresponding effect estimates remained significant (Supplementary Table 14). In addition, MR remained significant when restricted to the 56 signals that retained GWAS significance in meta-analysis (Supplementary Table 14). Conversely, MR also indicated that risk of schizophrenia is causal for longer sleep duration (inverse variance weighted: 0.025 min (0.007) per log odds ratio, $P = 6.05 \times 10^{-4}$; two-sample MR: weighted median: 0.026 min (0.006) per log odds ratio, $P = 3.36 \times 10^{-5}$) (Fig. 4, Supplementary Table 14). These data suggest a 1.04 h longer sleep duration per doubling in risk of schizophrenia. No other causal links were identified in two-sample MR. Furthermore, using two-sample MR with data from the GIANT consortium[39] ($n = 339,224$) and DIAGRAM consortium[40] ($n = 26,488$ cases and $n = 83,964$ controls), we found no evidence of causal effects of altered sleep duration with BMI and type 2 diabetes (Supplementary Table 13, Supplementary Figure 7).

**Discussion**

This study expands our understanding of the genetic architecture of self-reported sleep duration, estimating SNP-based heritability at 9.8%, consistent with earlier reports[41]. We identified 76 independent loci beyond the two previously known loci (*PAX8* and *VRK2*[14,17,18]). The largest effect remains at the *PAX8* locus (2.44 min per allele), consistent with previous reports[14,17,18]. Whereas individual signals exerted more modest effects on average (~1.04 min per allele), the aggregate effect of risk alleles could exceed 20 min, which is comparable to other well-recognized factors influencing sleep duration, such as gender[42]. Our GWAS findings were largely consistent upon adjustment for known risk factors, including BMI; however, attenuated effects were seen for some loci with adjustment for insomnia, reflecting some overlap between these sleep characteristics.

In separate GWAS for short and long sleep duration, 13 additional independent variants were identified, and only the *PAX8* locus was shared across all 3 GWASs. Our distinct findings for short and long sleep suggest the possibility of some distinct underlying biological mechanisms. As all three sleep traits were correlated, however, we did not account for multiple testing as have been done in previous GWAS of multiple correlated traits[18]. Future larger studies will be necessary to test if these loci reflect partially distinct genetic effects on short or long sleep, or reflect differences in statistical power in these dichotomized analyses.

The CHARGE consortium (adults) and EAGLE consortium (children/adolescents) sleep duration GWAS studies represent the largest independent available studies for replication, but are considerably smaller than the UK Biobank discovery cohort which limits opportunities for adequate replication[43]. Indeed, we had limited power to replicate individual SNPs in the two replication cohorts: the CHARGE consortium study with ~1/10th the sample size of the UK Biobank provided <12.5% power for all SNPs that did not individually replicate, and the EAGLE consortium study with ~1/40th the sample size of the UK Biobank provided power <70% for all SNPs that did not individually replicate. Despite these limitations, 52 loci remained significant after meta-analysis of both adult studies and 56 loci remained significant after meta-analysis of all three studies, a substantial advance over knowledge from prior studies. Future replication is necessary when appropriate resources become available, such as the US Department of Veterans Affairs Million Veteran Program and the All of Us Research program.

Furthermore, we validated effects of the combined sleep duration GRS in both adults and children/adolescents, further supporting our findings. Consistency between findings from the

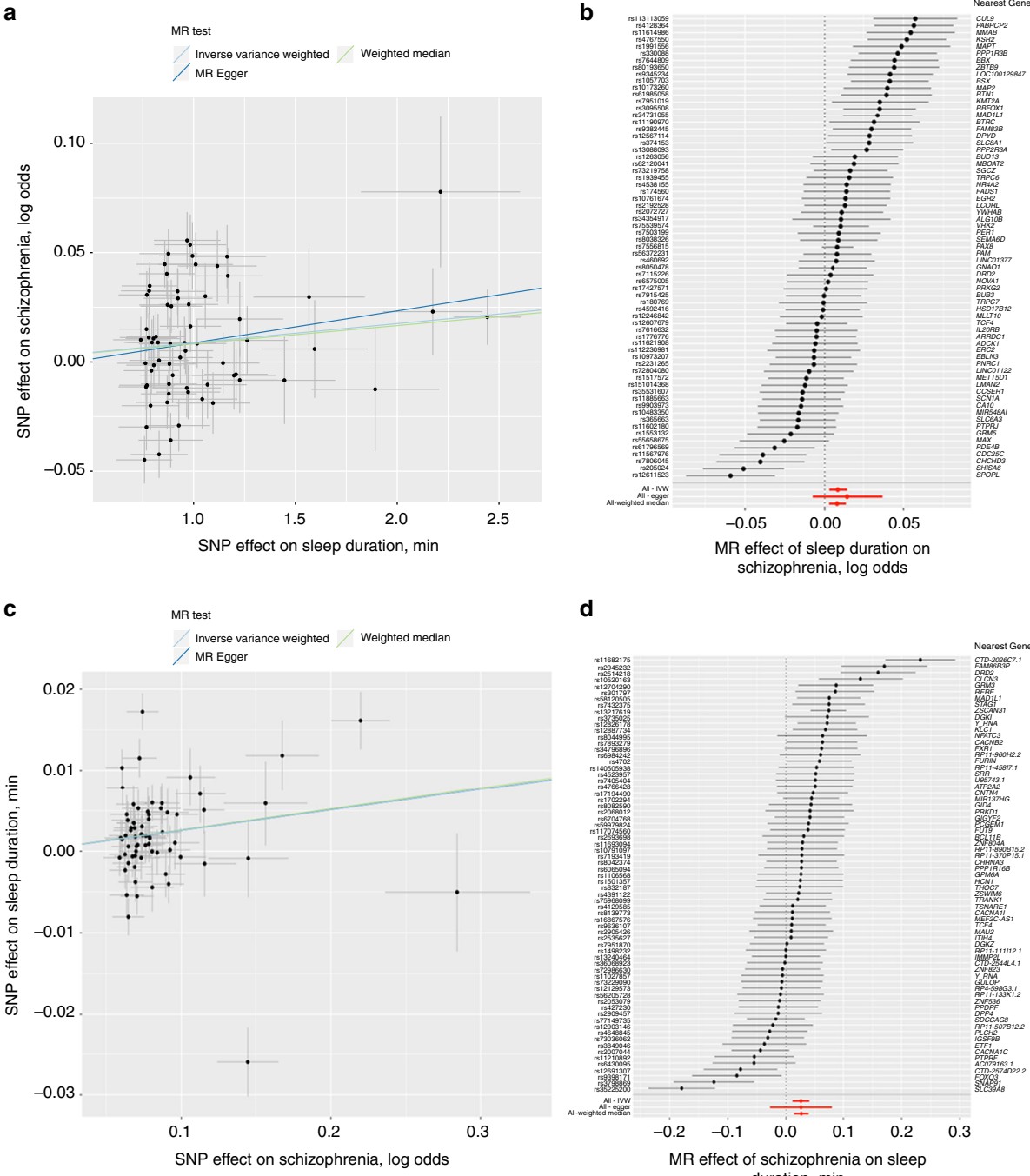

**Fig. 4** Bidirectional causal relationship of sleep duration with schizophrenia using Mendelian randomization. Association between single-nucleotide polymorphisms (SNPs) associated with sleep duration and schizophrenia (**a**) or SNP associated with schizophrenia and sleep duration (**c**) and forest plots show the estimate of the effect of genetically increased sleep duration on schizophrenia (**b**) or increased risk of schizophrenia on sleep duration (**d**). Lines identify the slopes for three Mendelian randomization (MR) association tests (**a**, **c**). Forest plots show each SNP with the 95% confidence interval (gray line segment; error bars) of the estimate and the inverse variance weighted, MR-Egger, and weighted median MR results in red

UK Biobank GWAS and sleep duration in independent studies despite differences in demographics and sleep duration ascertainment, two important extraneous factors that may influence self-report[44], reflect the generalizability of our signals. However, smaller effect estimates of the GRS in children/adolescents compared to adults supports previous studies that suggest the genetic architecture of sleep duration might differ between children and adults[19]. Furthermore, our finding of no significant overall genetic correlation between the GWAS of adults and children/adolescents sleep duration, as reported previously[19],

supports changes in sleep patterns throughout the lifespan[45–47], and larger GWAS of sleep duration in children/adolescents are needed.

Despite limitations of biases and imprecision in self-report, we observed largely consistent effects of our 78 signals with accelerometer-estimated sleep duration in a large subsample of 85,499 participants from the UK Biobank with up to 7 days of wrist-worn accelerometer. Self-report, actigraphy, and polysomnography estimated sleep duration provide both unique and overlapping information, have different sources of measurement

error, and may reflect different neurophysiological and psychological aspects[23,24,44]. Association of the sleep duration GRS with increased sleep efficiency, longer duration of daytime inactivity, but a larger number of sleep bouts, suggests that sleep duration genetic loci might impact other correlated parameters such as sleep latency, sleep fragmentation, and early awakening. Therefore, this secondary analysis allows us to begin exploring physiologic mechanisms underlying these associations. However, considering that the UK Biobank subsample with accelerometer data overlap with the discovery GWAS sample, these results should be interpreted with caution and further validation in an independent dataset is necessary. Furthermore, our study cannot resolve if a longer sleep GRS always reflects improved, higher quality sleep because of increased sleep efficiency, or may include qualitatively poorer, longer sleep or greater sleep need, given association with a larger number of sleep bouts and increased daytime napping. Thus, further investigation of the role of these loci in electroencephalography-derived physiological correlates of sleep architecture and sleep homeostasis from polysomnography and follow-up in cellular and animal models will help to dissect functional mechanisms.

We found compelling evidence of association near genes implicated with sleep traits in animal models, confirming that sleep–wake regulation is a highly conserved process with mechanisms shared between humans and model organisms. In agreement with the *FADS1/2* signal, we also observe an enrichment in genes related to unsaturated fatty acid metabolism, supporting experimental and observational evidence linking polyunsaturated fatty acids with sleep and related diseases, including neuropsychiatric disorders and depression[48–50]. We demonstrate enrichment in sleep duration GWAS signals within/near 80 genes identified as SNIPPs in mouse models[6], highlighting the potential importance of synaptic phosphorylation in sleep homeostasis in humans. A genetic variant, rs9382445, near the orexin receptor *HCRTR2* previously implicated for chronotype[17] associated significantly with sleep duration and retained significance and consistent effect estimates upon adjustment for diurnal preference.

Lastly, we extended the comparative analysis of the genetic architecture of sleep duration with other traits and found shared links between continuous sleep duration and cognitive, psychiatric, and disease traits, as well as u-shaped genetic correlations between short and long sleep duration and key lifestyle and disease traits including adiposity traits, years of schooling, and depressive symptoms. In bidirectional two-sample MR analyses, we observed causal links between longer sleep duration and increased risk of schizophrenia, consistent with previous findings[18,51], and conversely, causal links between schizophrenia and longer sleep duration. Associations remained robust in sensitivity analyses and the bidirectional causal link may suggest pleiotropy. Furthermore, MR effects may be biased due to collider bias as individuals with a genetic liability to neuropsychiatric diseases are underrepresented in studies such as the UK Biobank compared to the general population, while some independent protective factors for these conditions, including favorable sleep patterns, may be over-represented[52,53]. No other causal links were identified. Considering the non-specific and partially overlapping signals between the sleep duration and short/long sleep GWAS, we limited our MR analyses to sleep duration, and future MR should be carried out for short/long sleep duration separately. Follow-up MR analyses are warranted to verify null results identified in the one-sample insomnia MR, as well as other outcomes beyond those investigated.

In summary, our GWAS constitutes a large increase in associated loci for sleep duration that implicate multiple biological pathways and causal links to disease. This work and follow-up

studies will advance understanding of the molecular processes underlying sleep regulation and have the potential to identify new avenues of treatment for sleep and related disorders.

## Methods

**Population and study design**. Study participants were from the UK Biobank study, described in detail elsewhere[54]. In brief, the UK Biobank is a prospective study of >500,000 people living in the United Kingdom. All people in the National Health Service registry who were aged 40–69 years and living <25 miles from a study center were invited to participate between 2006 and 2010. In total, 503,325 participants were recruited from over 9.2 million invitations. Extensive phenotypic data were self-reported upon baseline assessment by participants using touchscreen tests and questionnaires and at nurse-led interviews. Anthropometric assessments were also conducted and health records were obtained from secondary care data from linked Hospital episode statistics (HES) obtained up until 04/2017. For the current analysis, 24,533 individuals of non-white ethnicity (as defined in genotyping and quality control) were excluded to avoid confounding effects. The UK Biobank study was approved by the National Health Service National Research Ethics Service (ref. 11/NW/0382), and all participants provided written informed consent to participate in the UK Biobank study.

**Sleep duration and covariate measures**. Study participants ($n \sim 500,000$) self-reported sleep duration at baseline assessment. Participants were asked: About how many hours sleep do you get in every 24 h? (please include naps), with responses in hour increments. Sleep duration was treated as a continuous variable and also categorized as either short (6 h or less), normal (7 or 8 h), or long (9 h or more) sleep duration. Extreme responses of less than 3 h or more than 18 h were excluded[17] and Do not know or Prefer not to answer responses were set to missing. Participants who self-reported any sleep medication (see Supplementary Method 1) were excluded. Furthermore, participants who self-reported any shift work or night shift work or those with prevalent chronic disease (i.e., breast, prostate, bowel or lung cancer, heart disease, or stroke) or psychiatric disorders (see Supplementary Method 2) were later additionally excluded in a secondary GWAS.

Participants further self-reported age, sex, caffeine intake (self-reported cups of tea per day and cups of coffee per day), daytime napping (*Do you have a nap during the day?*), smoking status, alcohol intake frequency (never, once/week, 2–3 times/week, 4–6 times/week, daily), chronotype (*Do you consider yourself to be …*, with the following response options: Definitely a 'morning' person, More a 'morning' than 'evening' person, More an 'evening' than a 'morning' person, and Definitely an 'evening' person), menopause status, and employment status during assessment. Socio-economic status was represented by the Townsend deprivation index based on national census data immediately preceding participation in the UK Biobank. Weight and height were measured and BMI was calculated as weight (kg)/height$^2$(m$^2$). Cases of sleep apnea were determined from self-report during nurse-led interviews or health records using International Classification of Diseases (ICD)-10 codes for sleep apnea (G47.3). Cases of insomnia were determined from self-report to the question, *Do you have trouble falling asleep at night or do you wake up in the middle of the night?* with responses never/rarely, sometimes, usually, prefer not to answer. Participants who responded *usually* were set as insomnia cases, and remaining participants were set as controls. Missing covariates were imputed using sex-specific median values for continuous variables (i.e., BMI, caffeine intake, alcohol intake, and Townsend index), or using a missing indicator approach for categorical variables (i.e., napping, smoking, menopause, employment, and chronotype).

**Activity-monitor-derived measures of sleep**. Actigraphy devices (Axivity AX3) were worn 2.8–9.7 years after study baseline by 103,711 individuals from the UK Biobank for up to 7 days[55]. Of these 103,711 individuals, we excluded 11,067 individuals based on accelerometer data quality. Samples were excluded if they satisfied at least one of the following conditions (see also http://biobank.ctsu.ox.ac. uk/crystal/label.cgi?id=1008): a non-zero or missing value in data field 90002 (Data problem indicator), good wear time flag (field 90015) set to 0 (No), good calibration flag (field 90016) set to 0 (No), calibrated on own data flag (field 90017) set to 0 (No), or overall wear duration (field 90051) less than 5 days. Additionally, samples with extreme values of mean sleep duration (<3 h or >12 h) or mean number of sleep periods (<5 or >30) were excluded. After non-white ethnicity exclusions, 85,502 samples remained. Sleep measures were derived by processing raw accelerometer data (.cwa). First, we converted .cwa files available from the UK Biobank to .wav files using Omconvert (https://github.com/digitalinteraction/ openmovement/tree/master/Software/AX3/omconvert) for signal calibration to gravitational acceleration[55,56] and interpolation[55]. The .wav files were processed with the R package GGIR to infer activity-monitor wear time[57], and extract the z-angle across 5-s time-series data for subsequent use in estimating the sleep period time window (SPT-window)[25] and sleep episodes within it[58].

The SPT-window was estimated using an algorithm[25] implemented in the GGIR R package and validated using polysomnography (PSG) in an external cohort consisting of 28 adult sleep clinic patients and 22 healthy good sleepers. Briefly, for each individual, median values of the absolute change in z-angle (representing the dorsal–ventral direction when the wrist is in the anatomical

position) across 5-min rolling windows were calculated across a 24 h period, chosen to make the algorithm insensitive to activity-monitor orientation. The 10th percentile was incorporated into the threshold distinguishing movement from non-movement. Bouts of inactivity lasting ≥30 min are recorded as inactivity bouts. Inactivity bouts that are <60 min apart are combined to form inactivity blocks. The start and end of longest block defines the start and end of the SPT-window[25].

Sleep duration: Sleep episodes within the SPT-window were defined as periods of at least 5 min with no change larger than 5° associated with the $z$-axis of the accelerometer[58]. The summed duration of all sleep episodes was used as indicator of sleep duration.

Sleep efficiency: This was calculated as sleep duration (defined above) divided by the time elapsed between the start of the first inactivity bout and the end of the last inactivity bout (which equals the SPT-window duration).

Number of sleep bouts within the SPT-window: This is defined as the number of sleep bouts separated by last least 5 min of wakefulness within the SPT-window. The least-active 5 h hours ($L5$) and the most-active 10 h ($M10$) of each day were defined using a 5 h and 10 h daily period of minimum and maximum activity, respectively. These periods were estimated using a rolling average of the respectively time window. L5 was defined as the number of hours elapsed from the previous midnight, whereas M10 was defined as the number of hours elapsed from the previous midday.

Sleep midpoint: Sleep midpoint was calculated for each sleep period as the midpoint between the start of the first detected sleep episode and the end of the last sleep episode used to define the overall SPT-window (above). This variable is represented as the number of hours from the previous midnight, e.g., 2am = 26.

Daytime inactivity duration: Daytime inactivity duration is the total daily duration of estimated bouts of inactivity that fall outside of the SPT-window. A minimum of 16 wear-hours was required for each night to be included. For non-wear data, the sleep phenotypes were imputed. Briefly, a minimum of 16 wear-hours was required for each night to be included. For each 15-min block that was classified as non-wear, data were replaced by the average of blocks at the same time periods from the other days in each individual record[57]. All activity-monitor phenotypes were adjusted for age at accelerometer wear, sex, season of wear, release (categorical; UK BiLeVe, UK Biobank Axiom interim, release UK Biobank Axiom full release), and number of valid recorded nights (or days for M10) when performing the association test in BOLT-LMM. Genetic risk scores for sleep duration, short sleep, and long sleep were tested using the weighted genetic risk score calculated by summing the products of the sleep trait risk allele count for all 78, 27, or 8 genome-wide significant SNPs multiplied by the scaled effect from the primary GWAS using the GTX package in R.

**Genotyping and quality control.** Phenotype data are available for 502,631 subjects in the UK Biobank. Genotyping was performed by the UK Biobank, and genotyping, quality control, and imputation procedures are described in detail here[59]. In brief, the following was conducted by the UK Biobank. Blood, saliva, and urine was collected from participants, and DNA was extracted from the buffy coat samples. Participant DNA was genotyped on two arrays, UK BiLEVE and UK Biobank Axiom with >95% common content and genotypes for ~800,000 autosomal SNPs were imputed to two reference panels. Genotypes were called using Affymetrix Power Tools software. Sample and SNPs for quality control were selected from a set of 489,212 samples across 812,428 unique markers. Sample quality control (QC) was conducted using 605,876 high-quality autosomal markers. Samples were removed for high missingness or heterozygosity (968 samples) and sex chromosome abnormalities (652 samples). Genotypes for 488,377 samples passed sample QC (~99.9% of total samples). Marker-based QC measures were tested in the European ancestry subset ($n = 463,844$), which was identified based on principal components of ancestry. SNPs were tested for batch effects (197 SNPs/batch), plate effects (284 SNPs/batch), Hardy–Weinberg equilibrium (572 SNPs/batch), sex effects (45 SNPs/batch), array effects (5417 SNPs), and discordance across control replicates (622 on UK BiLEVE Axiom array and 632 UK Biobank Axiom array) ($p$ value < $10^{-12}$ or <95% for all tests). For each batch (106 batches total) markers that failed at least one test were set to missing. Before imputation, 805,426 SNPs pass QC in at least one batch (>99% of the array content). Population structure was captured by principal component analysis on the samples using a subset of high-quality (missingness < 1.5%), high-frequency SNPs (>2.5%) (~100,000 SNPs) and identified the subsample of white British descent. In addition to the calculated population structure by the UK Biobank, we locally further clustered subjects into four ancestry clusters using $K$-means clustering on the principal components, identifying 453,964 subjects of European ancestry. The UK Biobank centrally further imputed autosomal SNPs to UK10K haplotype, 1000 Genomes Phase 3, and Haplotype Reference Consortium (HRC) with the current analysis using only those SNPs imputed to the HRC reference panel. Autosomal SNPs were pre-phased using SHAPEIT3 and imputed using IMPUTE4. In total, ~96 million SNPs were imputed. Related individuals were identified by estimating kinship coefficients for all pairs of samples, using only markers weakly informative of ancestral background. In total, there are 107,162 related pairs comprising 147,731 individuals related to at least one other participants in the UK Biobank.

**Genome-wide association analysis.** Genetic association analysis was performed in related subjects of European ancestry ($n = 446,118$) using BOLT-LMM[60] linear

mixed models and an additive genetic model adjusted for age, sex, 10 principal components of ancestry, genotyping array, and genetic correlation matrix [jl2] with a maximum per SNP missingness of 10% and per sample missingness of 40%. We used a genome-wide significance threshold of $5 \times 10^{-8}$ for each GWAS. Odds ratio (OR; 95% CI) estimates for short/long sleep are from adjusted PLINK[61] logistic regression analyses where genetic association analysis was also performed in unrelated subjects of white British ancestry ($n = 326,224$) using PLINK logistic regression and an additive genetic model adjusted for age, sex, 10 PCs, and genotyping array to determine SNP effects on sleep traits. We used a hard-call genotype threshold of 0.1, SNP imputation quality threshold of 0.80, and a minor allele frequency (MAF) threshold of 0.001. Genetic association analysis for the X chromosome was performed using the genotyped markers on the X chromosome with the additional –sex flag in PLINK. Similarly, sex-specific GWASs were also performed using BOLT-LMM[60] linear mixed models. Trait heritability was calculated as the proportion of trait variance due to additive genetic factors measured in this study using BOLT-REML[60], to leverage the power of raw genotype data together with low-frequency variants (MAF ≥ 0.001). Lambda inflation (λ) values were calculated using GenABEL in R, and estimated values were consistent with those estimated for other highly polygenic complex traits. Additional independent risk loci were identified using the approximate conditional and joint association method implemented in GCTA (GCTA-COJO)[62].

**Sensitivity analyses of top signals.** Follow-up analyses on genome-wide significant loci in the primary analyses included covariate sensitivity analyses adjusting for BMI, insomnia (continuous only), chronotype (continuous only), or caffeine intake adjustments individually, or a combined adjustment for lifestyle and clinical traits, including day naps, Townsend index, smoking, alcohol intake, menopause status, employment status, and sleep apnea in addition to baseline adjustments for age, sex, 10 principal components of ancestry, and genotyping array. Sensitivity analyses were performed using BOLT-LMM[60] linear mixed models using the same input set of SNPS (i.e., hard-call genotypes) as for the main GWAS, and OR (95% CI) estimates for short/long sleep are from adjusted PLINK[61] logistic regression analyses in unrelated subjects of white British ancestry.

**Replication and meta-analyses of sleep duration loci.** Using publicly available databases, we conducted a lookup of lead self-reported sleep duration signals in self-reported sleep duration GWAS results from adult (CHARGE; $n = 47,180$) and childhood/adolescent (EAGLE; $n = 10,554$). If lead signal was unavailable, a proxy SNP was used instead. As different imputation panels were compared to the UK Biobank, 8 of the 78 SNPs and 1 of the 78 SNPs were not covered in the CHARGE consortium and EAGLE consortium, respectively. In addition, we combined self-reported sleep duration GWAS results from adult (CHARGE) and childhood/adolescent (EAGLE) with the UK Biobank (primary model) in fixed-effects meta-analyses using the inverse variance weighted method in METAL[63]. Meta-analyses were conducted first separately (UK Biobank + CHARGE ($n = 3,044,490$ variants) or UK Biobank + EAGLE ($n = 7,147,509$ variants)), then combined (UK Biobank + CHARGE + EAGLE; $n = 2,545,157$ variants). A genetic risk score (GRS) for sleep duration was tested using the weighted GRS calculated by summing the products of the sleep duration risk allele count for as many available SNPs of the 78 genome-wide significant SNPs in each study (70 for CHARGE, 77 for EAGLE) multiplied by the scaled effect from the primary GWAS using the GTX package in R[64].

**Gene, pathway, and tissue enrichment analyses.** Genes overlapping the LD interval of the identified loci, defined by the furthest SNPs in a 1 Mb window with $r^2 > 0.05$, were identified by SNPsea[26]. Gene-based analysis was performed using Pascal[31]. Pascal gene-set enrichment analysis used 1077 pathways from KEGG, REACTOME, BIOCARTA databases, and a significance threshold was set after Bonferroni correction accounting for 1077 pathways tested ($P < 0.05/1,077$). Pathway analysis was also conducted using MAGMA[30] gene-set analysis in FUMA[65], which uses the full distribution of SNP $P$ values and is performed for curated gene sets and Gene Ontology (GO) terms obtained from MsigDB (total of 10,891 pathways). A significance threshold was set after Bonferroni correction accounting for all pathways tested ($P < 0.05/10,891$). Using Pascal, we created a custom pathway of the SNIPP genes[6] using human orthologs identified in DAVID (Database for Annotation, Visualization and Integrated Discovery; 79 out of 80 identified SNIPPs). We then verified enrichment of the pathway in our sleep duration GWAS (continuous, short, and long sleep). Tissue enrichment analysis was conducted using FUMA[65] for 53 tissue types, and a significance threshold was set following Bonferroni correction accounting for all tested tissues ($P < 0.05/53$). Single cell enrichment analysis was conducted in FUMA[65] utilizing the Tabula Muris[33] dataset, and a significance threshold was set following Bonferroni correction accounting for all tested cell types ($P < 0.05/115$). Integration of gene expression data with GWAS using transcriptome-wide association analyses in 11 tissues[34] identified 38 genes for which sleep duration SNPs influence gene expression in the tissues of interest (Supplementary Table 28). Integrative transcriptome-wide association analyses with GWAS were performed using the FUSION TWAS package[34] with weights generated from gene expression in 9 brain regions and 2 tissues from the GTEx consortium (v6). Tissues for TWAS testing

were selected from the FUMA tissue enrichment analyses and here we present significant results that survive Bonferroni correction for the number of genes tested per tissue and for all 11 tissues.

**Genetic correlation analyses.** Post-GWAS genome-wide genetic correlation analysis of LD Score Regression (LDSC)[66–68] using LDHub was conducted using all UK Biobank SNPs also found in HapMap3 and included publicly available data from 224 published genome-wide association studies, with a significance threshold after Bonferroni correction for all tests performed ($P < 0.05/224$ tests). LDSC estimates genetic correlation between two traits from summary statistics (ranging from $-1$ to 1) using the fact that the GWAS effect-size estimate for each SNP incorporates effects of all SNPs in LD with that SNP, SNPs with high LD have higher $X^2$ statistics than SNPs with low LD, and a similar relationship is observed when single study test statistics are replaced with the product of $z$-scores from two studies of traits with some correlation. Furthermore, genetic correlation is possible between case/control studies and quantitative traits, as well as within these trait types. We performed partitioning of heritability using the 8 pre-computed cell-type regions, and 25 pre-computed functional annotations available through LDSC, which were curated from large-scale robust datasets[66]. Enrichment both in the functional regions and in an expanded region ($+500$ bp) around each functional class was calculated in order to prevent the estimates from being biased upward by enrichment in nearby regions. The multiple testing threshold for the partitioning of heritability was determined using the conservative Bonferroni correction ($P < 0.05/25$ classes). Summary GWAS statistics will be made available at the UK Biobank web portal.

**Mendelian randomization analyses.** MR analysis was carried out using the R package MR-Base[69] (available: github.com/MRCIEU/TwoSampleMR), using the inverse variance weighted approach as our main analysis method[70], and MR-Egger[71] and weighted median estimation[72] as sensitivity analyses. MR results may be biased by horizontal pleiotropy—i.e., where the genetic variants that are robustly related to the exposure of interest (here sleep duration) independently influence levels of a causal risk factor for the outcome. Inverse variance weighted (IVW) assumes that there is either no horizontal pleiotropy, or that across all SNPs, horizontal pleiotropy is (i) uncorrelated with SNP-risk factor associations and (ii) has an average value of zero. MR-Egger assumes (i) but relaxes (ii) by explicitly estimating the non-zero mean pleiotropy, and adjusting the causal estimate accordingly. Estimation of the pleiotropy parameter means that the MR-Egger estimate is generally far less precise than the IVW estimate. The weighted median approach is valid if less than 50% of the weight is pleiotropic (i.e., no single SNP that contributes 50% of the weight or a number of SNPs that together contribute 50% should be invalid because of horizontal pleiotropy). Given these different assumptions, if all three methods are broadly consistent, this strengthens our causal inference. For all our MR analyses, except insomnia, we used two-sample MR, in which for all 78 GWAS hits identified in this study for sleep duration, we looked for the per allele difference in odds (binary outcomes) or means (continuous) with outcomes from summary publicly available data in the MR-Base platform. Results are therefore a measure of 'longer sleep duration' and sample 1 is UK Biobank (our GWAS) and sample 2 a number of different GWAS consortia covering the outcomes we explored. For interpretation purposes, inverse variance weighted MR causal effect estimates were converted to OR per hour by multiplying log ORs by 60 in order to represent the OR per hour, and then exponentiating. For significant two-sample MR findings, in sensitivity analyses, we further conducted leave-one-out analyses, MR using sleep duration effect estimates adjusting for standard confounders including BMI, insomnia, and other lifestyle factors, and restricted to those GWAS variants from each respective analysis, MR restricted to the signals that retained GWAS significance in meta-analysis, and lastly, reverse direction MR analysis. For reverse direction MR, inverse variance weighted MR causal effect estimates in minutes were first converted to hours by multiplying by 60, and then were converted by multiplying by 0.693 ($=\ln(2)$) in order to represent changes in sleep duration in hours per doubling in odds of the binary exposure. The number of SNPs used in each MR analysis varies by outcome because of some SNPs (or proxies for them) not being located in the outcome GWAS. In addition, for schizophrenia MR, we excluded two pleiotropic SNPs rs34556183 (near *HIST1H2BJ*) and rs13109404 (near *SLC39A8*) from our analysis and, to test for bidirectional links, we derived the schizophrenia instrumental variable using GWAS variants reported by the Psychiatric Genomics Consortium Schizophrenia working group[73].

## Data availability

Summary GWAS statistics are publicly available at the Sleep Disorder Knowledge Portal (http://sleepdisordergenetics.org/) and the UK Biobank website (http://biobank.ctsu.ox.ac.uk/).

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

## Acknowledgements

This research has been conducted using the UK Biobank Resource (UK Biobank application number 6818 and 9072). We would like to thank the participants and researchers from the UK Biobank who contributed or collected data. A.R.W. and T.M.F. are supported by 323195:SZ-245 50371-GLUCOSEGENES-FP7-IDEAS-ERC. B.E.C. is supported by K01-HL135405-01, R01-HL113338-04, R35-HL135818-01, American Thoracic Society Foundation Unrestricted Grant (Sleep). D.J.G. is supported by NIH/NHLBI contracts N01-HC-25195 and N02-HL-6-4278 and cooperative agreement U01HL53941. D.W.R. is supported by Wellcome Investigator award 107849/Z/15/Z. H.T. is supported by Dutch Medical Research Foundation grants (016.VICI.170.200 and VIDI 017.106.370). J.M.L. is supported by F32DK102323, 4T32HL007901. M.G. is supported by the Spanish Government of Investigation, Development and Innovation (SAF2017-84135-R) including FEDER co-funding, and NIDDK R01DK105072. M.K.R. is supported by The University of Manchester Research Infrastructure Fund. M.K.R. has acted as a consultant for GSK, Novo Nordisk, Roche and MSD, and also participated in advisory board meetings on their behalf. M.K.R. has received lecture fees from MSD and grant support from Novo Nordisk, MSD and GSK. M.N.W. is supported by Wellcome Trust Institutional Strategic Support Award (WT097835MF). S.E.J. is funded by the Medical Research Council (grant: MR/M005070/1). S.M.P. is funded by MH108908 (S.M.P.), HL135818 (S.R.). T.M.F. is funded by 323195:SZ-245 50371-GLUCOSEGENES-FP7-IDEAS-ERC. J.T. is funded by Diabetes Research and Wellness Foundation Fellowship. R.N.B. is funded by Wellcome Trust and Royal Society grant: 104150/Z/14/Z. H.S.D. and R.S. are supported by NIH R01DK107859, NIH R01 DK102696 (R.S., F.A.J.L.S.), MGH Research Scholar Fund (R.S.), and NIH R35-HL135818 (S.R.).

## Author contributions

The study was designed by H.S.D., S.E.J., J.M.L., V.T.v.H., D.A.L., M.K.R., M.N.W. and R.S. H.S.D., S.E.J., A.R.W., J.M.L., VT.v.H., J.A.R., Y.S., K.P., R.B., J.B., S.D.K., M.A.L., A.I. L., K.S., J.T., D.J.G., H.T., D.W.R., S.M.P., T.M.F., S.R., D.A.L., M.K.R., M.N.W. and R.S. participated in acquisition, analysis, and/or interpretation of data. H.S.D., J.M.L., H.W. and R.S. wrote the manuscript and all co-authors reviewed and edited the manuscript, before approving its submission. R.S. is the guarantor of this work and, as such, had full access to all the data in the study and takes responsibility for the integrity of the data and the accuracy of the data analysis.

## Additional information

**Competing interests:** F.A.J.L.S. has received speaker fees from Bayer Healthcare, Sentara Healthcare, Philips, Kellogg Company, and Vanda Pharmaceuticals. M.K.R. reports

receiving research funding from Novo Nordisk, consultancy fees from Novo Nordisk and Roche Diabetes Care, and modest owning of shares in GlaxoSmithKline. The remaining authors declare no competing interests.

Hassan S. Dashti [1,2], Samuel E. Jones [3], Andrew R. Wood[3], Jacqueline M. Lane[1,2,4], Vincent T. van Hees [5], Heming Wang [2,6,7], Jessica A. Rhodes[1,2], Yanwei Song [1,8], Krunal Patel[1,8], Simon G. Anderson [9], Robin N. Beaumont[3], David A. Bechtold [10], Jack Bowden[11,12], Brian E. Cade[2,6,7], Marta Garaulet[13,14], Simon D. Kyle[15], Max A. Little [16,17], Andrew S. Loudon[10], Annemarie I. Luik[15], Frank A.J.L. Scheer [2,7,18], Kai Spiegelhalder[19], Jessica Tyrrell [3], Daniel J. Gottlieb[6,7,20], Henning Tiemeier[21,22], David W. Ray [10], Shaun M. Purcell[23], Timothy M. Frayling[3], Susan Redline[24], Deborah A. Lawlor [11,12], Martin K. Rutter[10,25], Michael N. Weedon[3] & Richa Saxena [1,2,4]

[1]Center for Genomic Medicine, Massachusetts General Hospital and Harvard Medical School, Boston 02114 MA, USA. [2]Broad Institute, Cambridge 02142 MA, USA. [3]Genetics of Complex Traits, University of Exeter Medical School, Exeter EX2 5DW, UK. [4]Department of Anesthesia, Critical Care and Pain Medicine, Massachusetts General Hospital and Harvard Medical School, Boston 02114 MA, USA. [5]Netherlands eScience Center, Amsterdam 1098 XG, The Netherlands. [6]Division of Sleep and Circadian Disorders, Department of Medicine, Brigham and Women's Hospital, Boston 02115 MA, USA. [7]Division of Sleep Medicine, Harvard Medical School, Boston 02115 MA, USA. [8]Northeastern University College of Science, 176 Mugar Life Sciences, 360 Huntington Avenue, Boston, MA 02015, USA. [9]Division of Cardiovascular Sciences, School of Medical Sciences, Faculty of Biology, Medicine and Health, The University of Manchester, Manchester M13 9PL, UK. [10]Division of Endocrinology, Diabetes & Gastroenterology, School of Medical Sciences, Faculty of Biology, Medicine and Health, University of Manchester, Manchester M13 9PL, UK. [11]MRC Integrative Epidemiology Unit at the University of Bristol, Bristol BS8 2BN, UK. [12]Population Health Sciences, Bristol Medical School, University of Bristol, Bristol BS8 2BN, UK. [13]Department of Physiology, University of Murcia, Murcia 30100, Spain. [14]IMIB-Arrixaca, Murcia 30120, Spain. [15]Sleep and Circadian Neuroscience Institute, Nuffield Department of Clinical Neurosciences, University of Oxford, Oxford OX3 7LF, UK. [16]Department of Mathematics, Aston University, Birmingham B4 7ET, UK. [17]Media Lab, Massachusetts Institute of Technology, Cambridge 02139 MA, USA. [18]Medical Chronobiology Program, Division of Sleep and Circadian Disorders, Departments of Medicine and Neurology, Brigham and Women's Hospital, Boston 02115 MA, USA. [19]Clinic for Psychiatry and Psychotherapy, Medical Centre - University of Freiburg, Faculty of Medicine, University of Freiburg, Freiburg 79106, Germany. [20]VA Boston Healthcare System, Boston 02132 MA, USA. [21]Deprtment of Social and Behavioral Science, Harvard TH Chan School of Public Health, Boston 02115 MA, USA. [22]Department of Epidemiology, Erasmus Medical Center, Rotterdam 3015, The Netherlands. [23]Department of Psychiatry, Brigham & Women's Hospital, Harvard Medical School, 02115, Boston, MA, USA. [24]Departments of Medicine, Brigham and Women's Hospital and Beth Israel Deaconess Medical Center, Harvard Medical School, Boston 02115 MA, USA. [25]Manchester Diabetes Centre, Manchester University NHS Foundation Trust, Manchester Academic Health Science Centre, Manchester M13 9PL, UK. These authors contributed equally: Hassan S. Dashti, Samuel E. Jones, Andrew R. Wood. These authors jointly supervised this work: Martin K. Rutter, Michael N. Weedon, Richa Saxena.

