## [Peer review file · Nature Communications]

Reviewer #3 (Remarks to the Author):

This is a strong, detailed, and thoughtful manuscript that will make an important impact in advancing current understanding on the causes and consequences of sleep. The authors have clearly conducted significant additional efforts to address concerns from previous reviews. However, before I can recommend this manuscript for publication, I feel that following issues need to be addressed:

=== start major issues ===

1) I share reviewer 1's concern that the authors have no justification for reporting 78 variants in their manuscript.

The authors have countered by arguing that a combined genetic risk score of the 78 SNPs shows significant evidence of replication in the CHARGE consortium, thus "this replication provides further support for our SNP findings". To my mind this response doesn't add any further evidence for the claim of reporting 78 variants. Taking an extreme example, it would not surprise me if a genetic risk score constructed from all SNPs with $p < 5 \times 10^{-6}$ (100's of SNPs) would show even stronger evidence of replication. I propose that the authors clearly state in the manuscript how many of the individual SNPs actually replicate in the EAGLE + CHARGE consortia. The discussion section can then include the nicely worded text the authors used in response to reviewer 1 re: limitations of sample sizes in their replication data.

In addition, ~14.6M SNPs were considered for analysis. Adjustment for multiple testing here would indicate significance value of $p < 3.4 \times 10^{-9}$ (indicating just 36 variants associated with sleep duration in Table S2). Even if QC of those SNPs led to around ~8M SNPs in final analysis, this would indicate a significance value of $p < 6.25 \times 10^{-9}$ (42 variants). Such stricter criteria in the discovery GWAS would lead to a more stable set of variants in the sensitivity analysis (from eyeballing Table S3).

2) I have an issue with the use of an accelerometer subgroup of participants to test or confirm the discovery of lead variants from self-reported data in the entire group. This means that the same ~85k participants are used to i) partially discover self-report variants, and ii) test the effect of those discovered variants on acc measures.

The ideal scenario would be to conduct a discovery GWAS in ~360k UKBB participants who have self-report only data. Then self-report replication would take place in CHARGE and UKBB-accelerometer-

only participants, and accelerometer validation/replication would take place in an independent sample of participants. The authors should either conduct this re-analysis (I imagine all the bioinformatics tools are ready to go on this) ... or be very clear to label the accelerometer analysis as a sensitivity analysis, and also clearly highlight this as a limitation in the discussion text.

3) I propose the authors either remove the Mendelian Randomisation analysis, or check it's robustness via sensitivity checks. Such sensitivity checks would include: bidirectional analysis (does schizophrenia effect sleep duration?), leave one out analysis (to check sensitivity of causal inference to any individual genetic variant), and check to make sure that instrument variables are not associated with standard confounders. In addition, the authors should very clearly state whether two-sample MR is used (which is good) or not (which should be stated as a limitation). Currently some of the analyses are two sample (such as schizophrenia which doesn't include UKBB samples) and some is not (such as insomnia or BMI which do include UKBB samples, which would also have been used to identify your sleep duration instruments).

=== end major issues ===

=== start minor issues ===

To help the authors prepare their manuscript for publication, I also recommend the following minor issues be addressed:

1) "Prospective epidemiologic studies suggest that both short (<6,7 hours per night) and long (>8,9) hours per night) ..." Please pick one threshold value (i.e. 6 OR 7, but not both)

2) "The sleep duration association signals encompass >200 candidate causal genes ..." For easier reading, please state what analysis method or tool you used here.

3) In the genetic correlation paragraph in the results section, please explicitly state how many outcomes were significantly correlated with the exposure of interest (sleep duration).

4) Line 307, it would probably be good to reference/cite the DIAGRAM consortium paper.

5) Method, activity-monitor derived measures of sleep. For non-wear data are the sleep phenotypes imputed or not? If so, please state how this was done. If not, it would be worth noting this as a limitation.

6) Method, activity-monitor – please also explicitly state that your algorithm was validated using PSG in an external cohort of 28 sleep clinic patients and n=22 healthy good sleepers.

7) Method, genotyping and quality control – In this section it is unclear what the authors explicitly did to prepare SNPs for GWAS analysis, vs. what UKBB has already provided for all researchers. It would be good to clarify the text here.

8) Methods, sensitivity analyses of top signals. When using BOLT-LMM did you construct the linear mixed model using the same input set of SNPS (i.e. hard call genotypes) as for the main GWAS? If so, this is good and should be stated. If not, it should be acknowledged as a limitation as different underlying mixed models will have been created on which to base the sensitivity analysis.

=== end minor issues ===

Response to reviewer's comments

Reviewer #3 (Remarks to the Author):

This is a strong, detailed, and thoughtful manuscript that will make an important impact in advancing current understanding on the causes and consequences of sleep. The authors have clearly conducted significant additional efforts to address concerns from previous reviews. However, before I can recommend this manuscript for publication, I feel that following issues need to be addressed:

We would like to thank the referee for their constructive and thoughtful comments, which we address below. Reviewer comments are quoted in bold and our point-by-point responses follow. Modifications and additions to the manuscript are included in our responses to the reviewers and are highlighted in red in the revised manuscript.

== start major issues ==

1) I share reviewer 1's concern that the authors have no justification for reporting 78 variants in their manuscript.

The authors have countered by arguing that a combined genetic risk score of the 78 SNPs shows significant evidence of replication in the CHARGE consortium, thus "this replication provides further support for our SNP findings". To my mind this response doesn't add any further evidence for the claim of reporting 78 variants. Taking an extreme example, it would not surprise me if a genetic risk score constructed from all SNPs with $p < 5e-6$ (100's of SNPs) would show even stronger evidence of replication. I propose that the authors clearly state in the manuscript how many of the individual SNPs actually replicate in the EAGLE + CHARGE consortia. The discussion section can then include the nicely worded text the authors used in response to reviewer 1 re: limitations of sample sizes in their replication data.

In addition, ~14.6M SNPs were considered for analysis. Adjustment for multiple testing here would indicate significance value of $p < 3.4e-9$ (indicating just 36 variants associated with sleep duration in Table S2). Even if QC of those SNPs led to around ~8M SNPs in final analysis, this would indicate a significance value of $p < 6.25e-9$ (42 variants). Such stricter criteria in the discovery GWAS would lead to a more stable set of variants in the sensitivity analysis (from eyeballing Table S3).

In agreement with the Reviewer's comment, we removed the number "78" from the title, and now more clearly state in the manuscript how many of the individual SNPs replicate in the adult CHARGE GWAS (abstract) and associate in the meta-analysis combining all 3 studies (UK Biobank, CHARGE, and childhood/adolescent EAGLE). Furthermore, in the Discussion section, we have included a more extensive discussion on replication and limitations of sample sizes in the replication data, and referenced the recent commentary from this Journal regarding limitations in replication of results from mega biobanks (Huffman et al., *Nat Comm* 2018; ref 59).

The reviewer raises a second important issue in his/her comment. While the traditional GWAS threshold might arguably not be stringent enough for the number of tests performed, Bonferroni correction for the number of SNPs tested would be too conservative because extensive linkage disequilibrium in the genome of individuals of European ancestry makes many of these SNPs redundant with one another. To alert the reader, we now add a sentence in the abstract and results stating that 43 variants pass a more stringent threshold of $P < 6 \times 10^{-9}$ established using permutation testing for another sleep trait assessed in the UK Biobank (ref 34; accepted in *Nat Comm*), and independent replication of each variant is warranted. In addition, in the results, we consistently indicate how many variants achieve the most stringent threshold of $P < 6 \times 10^{-9}$ in the meta-analysis.

2) I have an issue with the use of an accelerometer subgroup of participants to test or confirm the discovery of lead variants from self-reported data in the entire group. This means that the same ~85k participants are used to i) partially discover self-report variants, and ii) test the effect of those discovered variants on acc measures.

The ideal scenario would be to conduct a discovery GWAS in ~360k UKBB participants who have self-report only data. Then self-report replication would take place in CHARGE and UKBB-accelerometer-only participants, and accelerometer validation/replication would take place in an independent sample of participants. The authors should either conduct this re-analysis (I imagine all the bioinformatics tools are ready to go on this) ... or be very clear to label the accelerometer analysis as a sensitivity analysis, and also clearly highlight this as a limitation in the discussion text.

We can understand the potential advantages of splitting the sample, but we have concerns that this would reduce statistical power to detect GWAS signals (Huffman et al., Nat Comm 2018; ref 59), especially given relatedness between the two UK Biobank subsets, and thereby diminish the excellent opportunity for discovery provided by the large UK Biobank cohort.

To avoid reducing the sample size of the discovery dataset, we follow the reviewer's alternative solution and explicitly label our accelerometer analysis as a secondary analysis of objective sleep in the Abstract and Manuscript, and highlight this limitation in the Discussion. We further emphasize that this secondary analysis allows us to begin to explore physiologic mechanisms by which genetic variants may influence habitual sleep duration. We trust that this will be acceptable to the reviewer and editorial team.

3) I propose the authors either remove the Mendelian Randomisation analysis, or check it's robustness via sensitivity checks. Such sensitivity checks would include: bidirectional analysis (does schizophrenia effect sleep duration?), leave one out analysis (to check sensitivity of causal inference to any individual genetic variant), and check to make sure that instrument variables are not associated with standard confounders. In addition, the authors should very clearly state whether two-sample MR is used (which is good) or not (which should be stated as a limitation). Currently some of the analyses are two sample (such as schizophrenia which doesn't include UKBB samples) and some is not (such as insomnia or BMI which do include UKBB samples, which would also have been used to identify your sleep duration instruments).

We assessed robustness of the Mendelian Randomization (MR) via sensitivity analyses as suggested by the Reviewer. Specifically, in order to refine our sleep duration-schizophrenia MR analysis, we first excluded two known pleiotropic SNPs rs34556183 (near *HIST1H2BJ*) and rs13109404 (near *SLC39A8*). Next, we re-ran MR using the remaining SNPs; in total, 67 SNPs present in both our GWAS and the Psychiatric Genomics Consortium schizophrenia GWAS were included in the MR analysis. All new MR results across all 3 models suggesting that longer sleep duration is causal for schizophrenia have become stronger and more significant compared to previously reported results [two-sample MR: weighted median: 0.008 (0.003) log OR per minute, $P = 3.95 \times 10^{-3}$; inverse variance weighted: 0.009 (0.003) log OR per minute, $P = 3.70 \times 10^{-3}$]. These data suggest that a one-hour longer sleep duration leads to a 61.6% increase risk for schizophrenia (derived by multiplying log ORs by 60 in order to represent the OR per hour, and then exponentiating). In leave-one-out sensitivity analyses, MR results remained robust and consistent (IVW: all $p < 0.0069$). Sensitivity MR analyses limited to signals from GWAS adjusting for confounders (BMI, insomnia, or other lifestyle traits) and using corresponding effect estimates remained significant. In addition, MR remained significant when restricted to the 56 signals that retained GWAS significance in meta-analysis. To assess for bidirectional link between sleep duration and schizophrenia, we conducted two-sample reverse MR and observed a causal relationship of increased risk of schizophrenia on longer sleep duration.

We have updated our MR results to include updated MR, leave-one-out analyses, other sensitivity analyses, and bidirectional results (Figure 4, Supplemental Table S30, S31, S32). In addition, we are now more explicit on whether one or two sample MR is used. We have excluded all one-sample MR analyses

whenever two-sample MR is possible for traits with available independent GWAS summary statistics. In the case of insomnia, we only report one-sample MR as no independent GWAS is available.

== **start minor issues** ==

To help the authors prepare their manuscript for publication, I also recommend the following minor issues be addressed:

1) “Prospective epidemiologic studies suggest that both short (<6,7 hours per night) and long (>8,9) hours per night) ...” Please pick one threshold value (i.e. 6 OR 7, but not both)

We have now specified <6 for short sleep and >9 for long sleep.

2) “The sleep duration association signals encompass >200 candidate causal genes ...” For easier reading, please state what analysis method or tool you used here.

We have now clarified in the Results and Method section the tool used to identify the relevant genes.

3) In the genetic correlation paragraph in the results section, please explicitly state how many outcomes were significantly correlated with the exposure of interest (sleep duration).

We have now specified that 8 genome-wide correlations were observed for sleep duration.

4) Line 307, it would probably be good to reference/cite the DIAGRAM consortium paper.

We now reference both the GIANT and the DIAGRAM consortium papers.

5) Method, activity-monitor derived measures of sleep. For non-wear data are the sleep phenotypes imputed or not? If so, please state how this was done. If not, it would be worth noting this as a limitation.

Yes, for non-wear data, the sleep phenotypes were imputed. Briefly, a minimum of 16 wear-hours was required for each night to be included. For each 15-minute block that was classified as non-wear, data were replaced by the average of blocks at the same time periods from the other days in each individual record. The method is described in greater detail in reference 71.

6) Method, activity-monitor – please also explicitly state that your algorithm was validated using PSG in an external cohort of 28 sleep clinic patients and n=22 healthy good sleepers.

We have now explicitly stated in the Methods that the algorithm was indeed validated using PSG in an external cohort of 28 sleep clinic patients and n=22 healthy good sleepers.

7) Method, genotyping and quality control – In this section it is unclear what the authors explicitly did to prepare SNPs for GWAS analysis, vs. what UKBB has already provided for all researchers. It would be good to clarify the text here.

We now explicitly discriminate between the quality control performed centrally by the UK Biobank and that performed locally by the authors.

8) Methods, sensitivity analyses of top signals. When using BOLT-LMM did you construct the linear mixed model using the same input set of SNPs (i.e. hard call genotypes) as for the main GWAS? If so, this is good and should be stated. If not, it should be acknowledged as a limitation as different underlying mixed models will have been created on which to base the sensitivity analysis.

Yes, the same input set of SNPs were used as for the main GWAS. We have now explicitly mentioned this in the Methods.

== *end minor issues* ==

Reviewer #3 (Remarks to the Author):

The authors have addressed my concerns, and I support the publication of this manuscript.

In the final 'camera-ready' version I encourage the authors to double-check two MR-extrapolation calculations below:

1) There may be a small error in line 320 which states "These data suggest that a one-hour longer sleep duration leads to a 61.6% increase risk for schizophrenia." I've plugged the values from Supp Table 30 into a calculator and can't repeat this result -> $\exp(.009 \times 60) =$ odds ratio of 1.6926 ... I then don't see how this converts to a "61.6% increase risk". Perhaps I have missed something, and it would be good to add an extra line in the methods for how you did this.

2) Similarly in line 331, I couldn't see exactly how you moved from a 1.566 hours per log odds ratio to stating that "These data suggest a 1.1 hour longer sleep duration per doubling in risk of schizophrenia." Again adding a sentence into the methods section on how you calculated this would be good (as well as stating " $\ln(2)$ " rather than "0.693" (it took a while for my brain to register this).

REVIEWERS' COMMENTS:

Reviewer #3 (Remarks to the Author):

The authors have addressed my concerns, and I support the publication of this manuscript.

We would like to thank the referee for their important comments, which we address below. The Reviewer comments are quoted in bold and our point-by-point responses follow. Modifications and additions to the manuscript are included in our responses to the reviewers and appear in track changes in the revised manuscript.

In the final 'camera-ready' version I encourage the authors to double-check two MR-extrapolation calculations below:

1) There may be a small error in line 320 which states "These data suggest that a one-hour longer sleep duration leads to a 61.6% increase risk for schizophrenia.". I've plugged the values from Supp Table 30 into a calculator and can't repeat this result -> $\exp(.009 \times 60) =$ odds ratio of 1.6926 ... I then don't see how this converts to a "61.6% increase risk". Perhaps I have missed something, and it would be good to add an extra line in the methods for how you did this.

Indeed, we have erroneously used the weighted median effect estimates instead of the inverse variance weighted effect estimates, the method which serves as our primary analysis method. Thus, we now include the updated results as the following: $\exp(0.0088 \times 60) = 1.696$ or 69.6%. In agreement with the Reviewer, we now add an extra line in the methods for how we computed this odds ratio.

We now also acknowledge the possibility of biased MR effect estimates due to collider bias. We have added the following in the Discussion: MR effects may be biased due to collider bias as individuals with a genetic liability to neuropsychiatric diseases are underrepresented in studies such as the UK Biobank compared to the general population, whilst some independent protective factors for these conditions, including favorable sleep patterns, may be over-represented.

2) Similarly in line 331, I couldn't see exactly how you moved from a 1.566 hours per log odds ratio to stating that "These data suggest a 1.1 hour longer sleep duration per doubling in risk of schizophrenia." Again adding a sentence into the methods section on how you calculated this would be good (as well as stating "ln(2)" rather than "0.693" (it took a while for my brain to register this).

Similar to comment 1, we now update our calculations in order to use the inverse variance weighted effect estimates, the method which serves as our primary analysis method. We now include the updated results as the following: $\ln(2) \times (0.025 \times 60) = 1.04$ hours. In agreement with the Reviewer, we now add an extra line in the methods for how we computed this estimate.